# Flexible decapyrrylcorannulene hosts

Yun-Yan Xu[1], Han-Rui Tian[1], Shu-Hui Li[1], Zuo-Chang Chen[1], Yang-Rong Yao[1], Shan-Shan Wang[1], Xin Zhang[1], Zheng-Zhong Zhu[1], Shun-Liu Deng[1], Qianyan Zhang[1], Shangfeng Yang [2], Su-Yuan Xie[1], Rong-Bin Huang[1] & Lan-Sun Zheng[1]

The assembly of spherical fullerenes, or buckyballs, into single crystals for crystallographic identification often suffers from disordered arrangement. Here we show a chiral configuration of decapyrrylcorannulene that has a concave 'palm' of corannulene and ten flexible electron-rich pyrryl group 'fingers' to mimic the smart molecular 'hands' for self-adaptably cradling various buckyballs in a (+)hand-ball-hand(−) mode. As exemplified by crystallographic identification of 15 buckyball structures representing pristine, exohedral, endohedral, dimeric and hetero-derivatization, the pyrryl groups twist with varying dihedral angles to adjust the interaction between decapyrrylcorannulene and fullerene. The self-adaptable electron-rich pyrryl groups, susceptible to methylation, are theoretically revealed to contribute more than the bowl-shaped palm of the corannulene in holding buckyball structures. The generality of the present decapyrrylcorannulene host with flexible pyrryl groups facilitates the visualization of numerous unknown/unsolved fullerenes by crystallography and the assembly of the otherwise close-packed spherical fullerenes into two-dimensional layered structures by intercalation.

[1] State Key Lab for Physical Chemistry of Solid Surfaces, Collaborative Innovation Center of Chemistry for Energy Materials, Department of Chemistry, College of Chemistry and Chemical Engineering, Xiamen University, Xiamen 361005, China. [2] Hefei National Laboratory for Physical Sciences at Microscale, CAS Key Laboratory of Materials for Energy Conversion, Department of Materials Science and Engineering, Synergetic Innovation Center of Quantum Information & Quantum Physics, University of Science and Technology of China, Hefei 230026, China. These authors contributed equally: Yun-Yan Xu, Han-Rui Tian, Shu-Hui Li. Correspondence and requests for materials should be addressed to Q.Z. (email: xmuzhangqy@xmu.edu.cn) or to S.Y. (email: sfyang@ustc.edu.cn) or to S.-Y.X. (email: syxie@xmu.edu.cn)

Fullerenes, also known as cage-like 'buckyballs', represent one class of the most remarkable molecules discovered in the last three decades[1]. However, a number of examples have been retracted due to problematical structural determinations that are largely attributed to the lack of reliable analytical approaches[2–4]. Crystallography is regarded as the most reliable approach towards the unambiguous identification of molecular structures, but the presence of severe disorder defects in the single crystals of spherical fullerenes frequently leads to ambiguous geometric identification. Exohedral derivatization has been employed to reduce the disorder defects, as exemplified by determination of the $C_{60}$ cage structure through the crystallographic study of its derivative obtained from a cycloaddition reaction involving $OsO_4$ and 4-tert-butylpyridine[5]. The exohedral derivatization method is even adequate to crystallographically identify some of the well-known fullerene derivatives, such as α-$PC_{71}BM$[6], a dominant isomer of the most prevalent electron acceptor [6,6]-phenyl-$C_{71}$ butyric acid methyl ester. Alternatively, the supramolecular assembly of buckyballs into co-crystals with a selected 'buckycatcher' host is also a straightforward approach for reducing the disorder defects and identifying the pristine structures of various fullerenes[7–14]. In buckycatcher/buckyball (host/guest) systems, that are generally applied in a stoichiometric ratio of 1:1, the most prevalent buckycatchers are porphyrin compounds such as $M^{II}$ (OEP) (OEP=octaethylporphyrin)[15–17]. Tens of fullerenes, especially endohedral ones[18], have been identified in the form of supramolecular $M^{II}$(OEP)/fullerene co-crystals since the groundbreaking work by Balch and co-workers in 1999[15].

In addition to $M^{II}$(OEP), concave corannulene[19] is an ideal host for fullerenes with convex surfaces. Scott and co-workers first reported the assembly of $C_{60}$ with a penta-tert-butyl corannulene derivative[20] in the solid state and a corannulene polysulfide[21] in solution, illustrating that the attachment of adaptable substituents to the periphery of corannulene is an effective way to tune the electron distribution of corannulene and consequently reinforce the anticipated attraction between corannulene and fullerenes. Similarly, the supramolecular assemblies formed between our recently synthesized hexathiolated trithiasumanenes[22] and fullerenes demonstrate the importance of functionalization with electron-rich substituents. Instead of pristine corannulene, doping heteroatoms in corannulene can also enhance its electron-donating and polarizing abilities, leading to stronger supramolecular interactions between heteroatom-doped corannulene[12] and fullerenes. On the other hand, an increase in surface contact between the host and guest molecules also favours the formation of assemblies. Recent computations revealed that the binding energy of a bowl-shaped poly-arene and buckyball was proportional to the surface contact area[23]. Indeed, both dibenzocorannulene[24] and double concave biscor-annulenes[7] have enhanced associations with fullerenes due to their expanded concave surfaces, but the resultant rigid cavities limit the compatibility of these derivatives to self-adaptly interact with full-erenes upon changing shapes/types of buckyballs.

Herein, we design and synthesize a unique decapyrrylcor-annulene (abbreviated as DPC) that mimics a molecular 'hand', where ten pyrryl groups resemble flexible 'fingers' on the per-iphery of a bowl-shaped corannulene 'palm'. Benefiting from the matched shape, electron-rich property, expanded surface contact and flexible pyrrole-corannulene dihedral angles, the DPC host is capable of holding almost all commonly known types of fullerene such as pristine, exohedral, endohedral, dimeric and hetero-derivatized structure as well as fulleroid and pentagon-fused fullerene.

## Results and discussion

### Synthesis and structures of decapyrrylcorannulene derivatives.

The syntheses of DPC (**2a**) and methyl-substituted DPC [decakis (3,4-dimethylpyrryl)corannulene, **2b**] were accomplished by a facile metal-free catalytic Ullmann reaction under mild condi-tions, as shown in Fig. 1. Inspired by the successful synthesis of corannulene polysulfide from decachlorocorannulene reported by Sigel et al.[25]. (Supplementary Note 1) and hexapyrrylbenzene from hexafluorobenzene reported by Henri[26], Meijer[27]and Muellen et al.[28], we conducted a reaction using pyrryl sodium with decachlorocorannulene **1** at 25 °C in DMF (Supplementary Notes 2-3). Pure **2a** and **2b** were obtained as yellow solids with good solubilities in commonly used solvents, such as dichlor-omethane, carbon disulfide, benzene and toluene.

The molecular structures of **2a** and **2b** were established by nuclear magnetic resonance (NMR) spectroscopy (Supplementary Figs. 1–4) and mass spectrometry. The photophysical properties of **2a** were investigated as shown in Supplementary Figs. 7, 8, Supplementary Note 6 and Supplementary Table 1. The bowl-shaped geometric structures of **2** were unambiguously identified by single-crystal X-ray diffraction of the yellow rod-shaped crystals that were obtained by slow solvent evaporation from

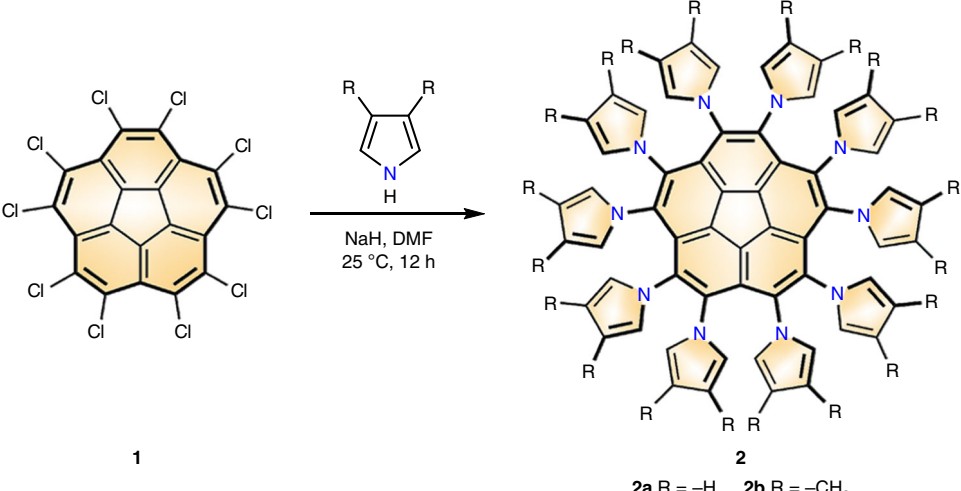

**Fig. 1** Synthesis of decapyrrylcorannulene derivatives 2 from decachlorocorannulene 1. Conditions: NaH (60% oil dispersion, 2.0 mmol, 12 eq for **2a**, 3.4 mmol, 20 eq for **2b**), pyrrole (2.0 mmol, 12 eq) or 3,4-dimethylpyrrol (3.4 mmol, 20 eq), decachlorocorannulene **1** (0.17 mmol, 1 eq), 25 °C, 12 h. DMF: N, N-dimethylformamide

dichloromethane solution of **2a** or **2b**. As shown in Fig. 2a, c, the 10 pyrryl groups all twist in the same direction but with varying dihedral angles (Supplementary Fig. 9 and Supplementary Table 3), which gives **2a** and **2b** molecules chiral configurations in the crystals. This chirality derives from *P* vs *M* twisting of the 10 pyrryl groups in the crystal, rather than the concave vs convex geometry of the corannulene core[29]. With the attachment of pyrryl groups to the periphery of corannulene, the bowl depth was reduced from the value of 0.87 Å reported for parent corannulene[29] to 0.61 Å for **2a** and 0.60 Å for **2b**, indicating a decrease in the corannulene curvature, which is possibly due to the repulsion of the 10 pyrryl groups (Fig. 2a, c, Supplementary Fig. 18 and Supplementary Table 11), similar to the results shown for decaphenylcorannulene[30]. However, based on the distance from the mean plane of the centre corannulene core pentagon to the α-carbon of the pyrrole ring, the concave surfaces of **2a** and **2b** actually expanded relative to that of parent corannulene, and, in some senses, the concave depth increased to 1.97 Å for **2a** and 1.96 Å for **2b**, as shown in Fig. 2a, c. The inversion barrier energy of **2a** was calculated to be 3.11 kcal mol$^{-1}$ by density functional theory (DFT) calculations, a value much lower than those of pristine corannulene (10.46 kcal mol$^{-1}$) and sumanene (19.29 kcal mol$^{-1}$) calculated at the same level of theory (see Supplementary Table 12 and 15).

Usually, buckybowls tend to stack into a columnar supramolecular structure in a concave-to-convex orientation[31,32], and few cases of structures that differ from columnar packing have been reported[33]. Unexpectedly, **2a** exhibits an alternate convex-to-convex plus concave-to-concave columnar stacking structure along the *a* axis, as shown in Fig. 2b. The stacking distance is 3.34 Å between neighbouring convex-to-convex faces and 7.36 Å between concave-to-concave faces, which form untethered clam-like cavities, implying that fullerene $C_{60}$ with a diameter of ~7 Å may be allowed to intercalate into the untethered clam-like cavity. In addition to the unique concave-to-concave packing

orientation, a layer-by-layer structure is observed in the crystal, as shown in the *a*–*c* plane of the crystal depicted in Fig. 2b. Such an exceptional packing pattern might be mainly due to the influence of the flexible pyrryl groups, which afford multiple self-adaptable C-H···π interactions (edge to face) in both the convex-to-convex and concave-to-concave packing orientations. Note that the generation of these significant C-H···π interactions mainly depends on the ability of the flexible pyrryl groups to adopt self-adaptive and suitable dihedral angles (ranging from 62.27° to 66.80°, as seen in Supplementary Table 3). Methyl-substituted DPC (**2b**) adopts a completely different packing mode than **2a**, as shown in Fig. 2d. With the introduction of methyl groups, the previous convex-to-convex stacking orientation of **2a** still exists in the packing structure of **2b**, but the unique concave-to-concave stacking mode is transformed into a more compact cross-orthogonal packing pattern (see Supplementary Fig. 19 and Supplementary Data 1), with implication about **2b** unsuitable for forming supramolecular assemblies with fullerenes in the crystal state.

**Supramolecular behaviours of the DPC host/buckyball guest systems.** The structure of **2a** featuring an expanded concave depth (1.97 Å) matches well with the convex structure of the fullerene surface, implying that the formation of a supramolecular assembly between **2a** and fullerenes is possible. DFT calculations predicted a higher binding energy for the assembly of **2a** with $C_{60}$ than for the assembly of pristine corannulene with $C_{60}$ (31.39–31.44 vs 15.46 kcal mol$^{-1}$) (see Supplementary Table 13). Indeed, **2a** undergoes strong supramolecular assembly with $C_{60}$ and $C_{70}$ to form **2a**/$C_{60}$ and **2a**/$C_{70}$ supramolecules in a stoichiometric ratio of 1:1 in solution on the basis of the $^1$H NMR complexation-induced chemical shifts[21] (see Supplementary Fig. 15 and Supplementary Note 7). Average association constants ($K_a$) of 8252 and 5686 M$^{-1}$ were determined for the

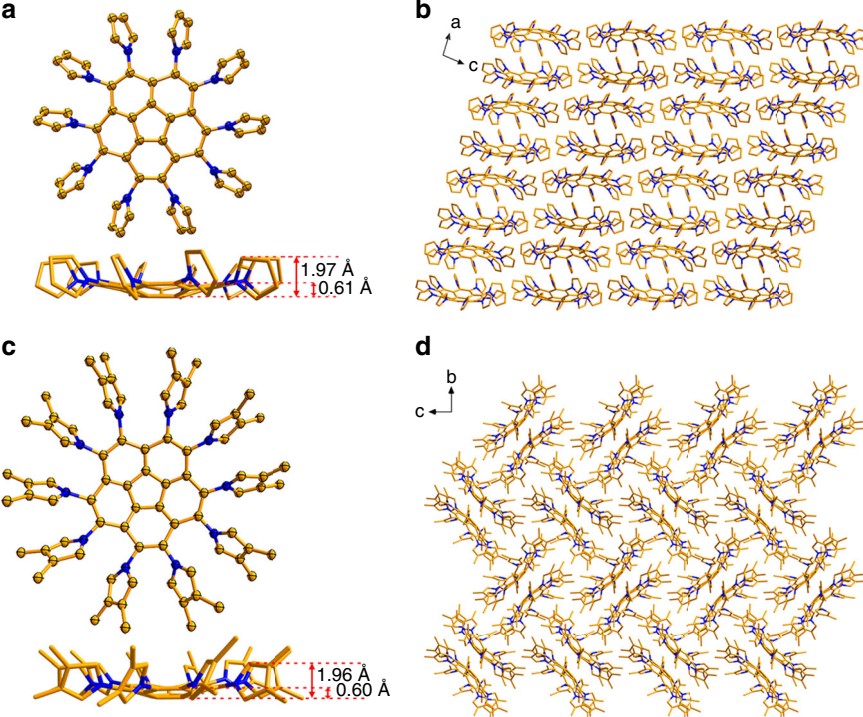

**Fig. 2** Crystallographic structures and packing of **2a** and **2b**. **a**, **c** Crystallographic structures of **2a** and **2b** with thermal ellipsoids at the 50% probability level. The bowl depths of **2a** and **2b** are indicated. **b**, **d** Packing structures of **2a** in the *a*–*c* plane and **2b** in the *b*–*c* plane. C atom is shown in gold and N atom in navy blue. All hydrogen atoms are omitted for clarity

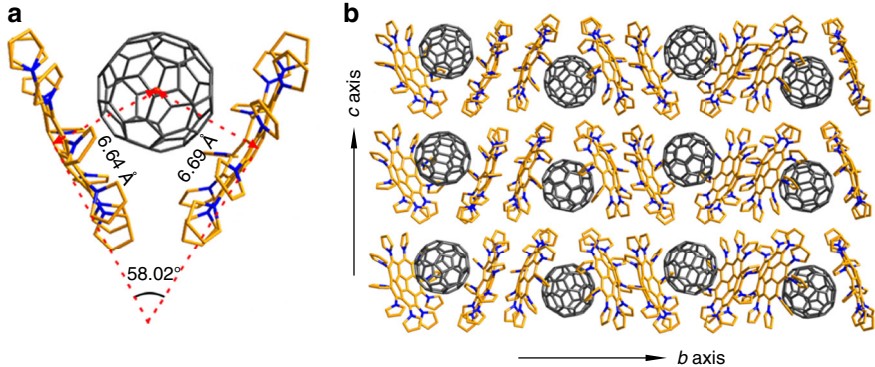

**Fig. 3** Crystallographic structure and packing of 2DPC{$C_{60}$}. **a** Side view of 2DPC{$C_{60}$}. **b** Packing structure of 2DPC{$C_{60}$} in the $b$–$c$ plane. C atom in **2a** is shown in gold, $C_{60}$ in grey and N atom in navy blue. All hydrogen atoms and solvent molecules ($CS_2$) are omitted for clarity

supramolecules **2a**/$C_{60}$ and **2a**/$C_{70}$, respectively, in toluene-$d_8$ at room temperature by nonlinear curve flitting of the NMR data (see Supplementary Figs. 12–14). These association constants are comparable to that of the double concave 'buckycatcher' (8600 M$^{-1}$) in toluene-$d_8$ reported by Sygula[7], clearly indicating the strong binding interaction of **2a** with fullerenes. Additionally, the formation of supramolecules in solution was also confirmed by the observation of fluorescence quenching when $C_{60}$ was titrated into solutions of **2a** (as shown in Supplementary Fig. 16). Such strong supramolecular interactions are also attributed to the introduction of electron-rich pyrryl groups on the periphery of corannulene. By contrast, the previously reported corannulene derivatives, pentakis(tert-butyl)corannulene[20] and dibenzo-[a,g]corannulene[24] were unable to form supramolecular assemblies with $C_{60}$ in solution.

The supramolecular assembly behaviour of **2a** with fullerenes in the solid state can be characterized by X-ray crystallography (see Supplementary Tables 4–8). With the introduction of $C_{60}$, as shown in Fig. 3a, the clam-like cavities formed by the concave-to-concave stacking orientation of **2a** are occupied by $C_{60}$ molecules in the co-crystal of **2a**/$C_{60}$. The shell of the clam-like cavity cooperatively opens into a V shape with an angle of 58.02° (Fig. 3a). Upon the involvement of $C_{60}$, the dihedral angles of pyrryl group and corannulene rim in **2a**/$C_{60}$ co-crystal vary from 57.07° to 68.61° (see Supplementary Table 9), the variation range is larger than the span of 62.27°−66.80° observed in the crystal of pristine **2a** (see Supplementary Table 3), reflecting the pyrryl groups are adaptable and capable of adjusting the interactions between **2a** and $C_{60}$. The asymmetric unit consists of one pair of fully ordered **2a** molecules and one molecule of $C_{60}$, which is disordered with two orientations. Interestingly, in each pair of **2a** molecules the 10 pyrryl rings in one **2a** molecule all twist in the opposite direction relative to those in another molecule with varying dihedral angles (Supplementary Table 9), which gives the **2a** molecules a pair of chiral hand-like configuration. Each pair of chiral **2a** molecules [marked as (+) and (−)] cradle a $C_{60}$ cage in a structure strongly resembling a ball held by two hands as shown in Fig. 3a. The notation 2DPC{$C_{60}$} is adopted to specify this unique supramolecular (+)hand-ball-hand(−) assembly in which the buckyball $C_{60}$ is cradled by both chiral DPC hosts. Based on the interesting concept reported in Balch's article[34], our DPC hosts provides a valuable opportunity to sort chiral fullerene enantiomers in the future.

Within the unit cell of 2DPC{$C_{60}$}, fullerene is positioned asymmetrically between the pair of **2a** molecules. Measured from the plane of the five hub carbons of **2a** to the centroid of $C_{60}$, the penetration depths of $C_{60}$ into (+) and (−) **2a** are 6.64 Å and 6.69 Å (Fig. 3a, Supplementary Table 10), which are shorter than those observed in the supramolecular corannulene/$C_{60}$ (6.94 Å)[20] and azabuckybowl/$C_{60}$ (6.82 Å)[12]. Such a short contact suggests the occurrence of strong π-π interactions between **2a** and $C_{60}$.

In the crystal packing structure of 2DPC{$C_{60}$}, the $C_{60}$ molecules are completely surrounded by the DPC hosts of **2a** and aggregate in a zigzag fashion along the $b$ axis, as shown in Fig. 3b. Along the $c$ axis, the DPC hosts and buckyballs stack head-to-tail, forming a one-dimensional badminton-shaped packing structure, and the adjacent columns pack with the opposite badminton shape. The π-π or C-H⋯π intermolecular contacts that exist between the DPC host and buckyball and between two DPC host molecules are tuneable through the dihedral angles of the pyrryl groups. In particular, the weak interactions present in the structure are quite sensitive for the formation of 2DPC{$C_{60}$} because another DPC derivative, decakis (3,4-dimethylpyrryl)corannulene **2b**, in which all (β)H atoms of the pyrryl groups are replaced by methyl groups, does not easily assemble with any fullerenes, as suggested in the crystal structure of **2b** (see Supplementary Fig. 19 and Supplementary Data 1). This difference in behaviour highlights the importance of the cooperative and adaptable weak intermolecular interactions, especially the appropriate C-H⋯π interactions, within the co-crystal of 2DPC{$C_{60}$}.

The merit of **2a** is embodied by the generality for holding any fullerene, regardless of the shape/type of fullerene derivative. As shown in Fig. 4, all the common types of fullerenes, including pristine ($C_{60}$, $C_{70}$, $C_{90}$), exohedral (six methanofullerene[35] derivatives, three fullerene hydride derivatives[36], and one fulleroid[37] derivative), endohedral (Sc$_3$N@C$_{80}$)[16], and dimeric/hetero- [(C$_{59}$N)$_2$][38] fullerenes, have been co-crystallized in the forms of 2DPC{fullerene} suitable for crystallographic identification. Specifically, six methanofullerene derivatives with various functional groups such as phenyl, pyridyl and ester groups are compatible with **2a** for co-crystallization (Fig. 4d–i). Although a large number of literatures have discussed the synthesis of fullerene hydride derivatives, the involved structures have rarely been identified by crystallography. Here the DPC host **2a** help to visualize the fullerene hydride derivatives, including an IPR-defying (IPR=Isolated Pentagon Rule[39]) $C_{65}H_6$ and two unsolved $C_{60}HCH_3$ and $C_{60}HPh$ (Fig. 4j–l). As shown in Fig. 4d, e, h, i, and m, in addition, isomeric PC$_{71}$BM (α- and β$_1$-PC$_{71}$BM) and $C_{71}H_2$ (isomers I-III, see Supplementary Note 4) can be successfully held by the DPC host **2a** no matter where the derivative groups are located in the skeleton of fullerene isomers. In particular, the fulleroid of $C_{2v}$-$C_{71}H_2$-III was synthesized in 2010[40] but whose crystal structure has not been solved heretofore. In the present work, we succeeded in obtaining a high-quality single co-crystal of 2DPC{$C_{2v}$-$C_{71}H_2$-III}, in addition to the newly isolated isomers

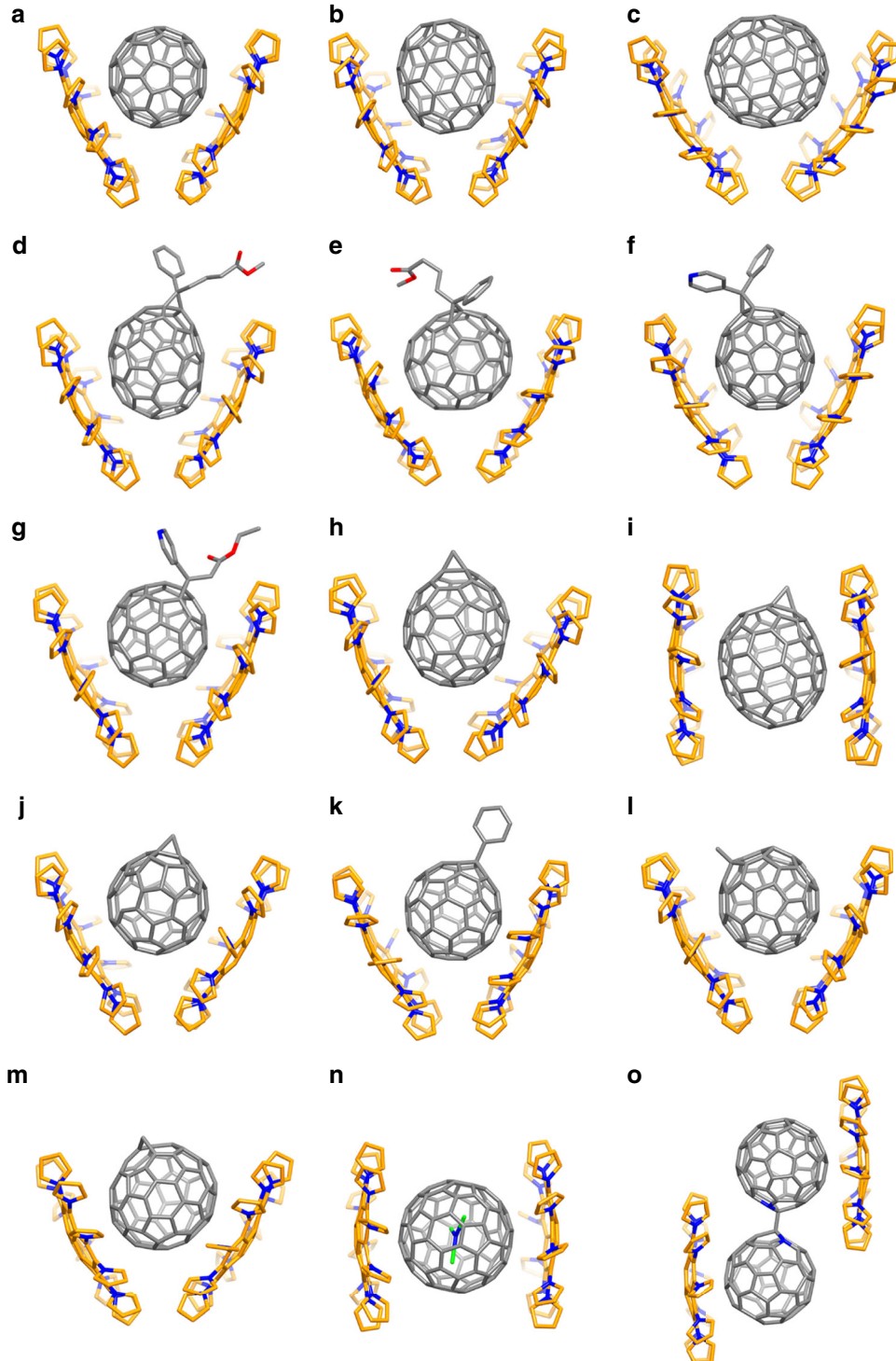

**Fig. 4** Crystallographic structures of co-crystals between DPC (**2a**) and fullerenes. **a** 2DPC{$C_{60}$}, **b** 2DPC{$C_{70}$}, **c** 2DPC{$C_{90}$}, **d** 2DPC{α-PC$_{71}$BM}, **e** 2DPC{β$_1$-PC$_{71}$BM}, **f** 2DPC{PC$_{61}$P}, **g** 2DPC{PC$_{61}$AE}, **h** 2DPC{$C_s$-C$_{71}$H$_2$-I}, **i** 2DPC{$C_s$-C$_{71}$H$_2$-II}, **j** 2DPC{C$_{65}$H$_6$}, **k** 2DPC{C$_{60}$HPh}, **l** 2DPC{C$_{60}$HCH$_3$}, **m** 2DPC{$C_{2v}$-C$_{71}$H$_2$-III}, **n** 2DPC{Sc$_3$N@C$_{80}$}, **o** 2DPC{(C$_{59}$N)$_2$}. PC$_{61}$P: [6,6]-phenyl-C$_{61}$-pyridyl. PC$_{61}$AE: [6,6]-pyridyl-C$_{61}$-acetic acid ethyl ester. C atom is shown in gold, N atom in navy blue, Sc atom in green, and O atom in red. All hydrogen atoms and solvent molecules in the crystal structures have been omitted for clarity

of 2DPC{$C_s$-C$_{71}$H$_2$-I} and 2DPC{$C_s$-C$_{71}$H$_2$-II} (see Supplementary Figs 5, 6 for HPLC separation of three C$_{71}$H$_2$ isomers from combustion soot[41] in details). The co-crystal structure of 2DPC{$C_{2v}$-C$_{71}$H$_2$-III} (Fig. 4m) reveals that the methylene group indeed stays on the equator of C$_{70}$, the same with the theoretical prediction[40]. Remarkably, the exact crystal structure of another

long-known pendent fullerene, (C$_{59}$N)$_2$ dimer, which was first reported by Wudl and co-workers in 1995[38] as the first heterofullerene dimer, has remained a mystery until now. Owing to the adaptability of DPC host **2a**, we succeeded in determining the molecular structure of this fullerene dimer via co-crystallization with **2a** for the first time (Fig. 4o and

Supplementary Tables 7–10). Our present result strongly supports the theoretical prediction by Lee et al.[42] that the gauche conformer is the lowest-energy structure, and the dihedral angle of deflection for two nitrogen atoms is approximately 65.57° in the co-crystal, slightly larger than that in the theoretical calculation[42].

The merit of **2a** is also reflected by the flexibility in cradling various buckyballs. Taking the pristine $C_{70}$, exohedral $\alpha$-$PC_{71}BM$, endohedral $Sc_3N@C_{80}$, fulleroid $C_{2v}$-$C_{71}H_2$-III, and dimeric/hetero-$(C_{59}N)_2$ as representative examples (Supplementary Table 9), the flexibility of the 2DPC{fullerene} assembly conformations is attributed to ten pyrryl groups of **2a** that are able to self-adjust the dihedral angles for matching different shape/type of fullerene. As shown, for most of supramolecular 2DPC{fullerene}, V-shaped conformations are found (Fig. 4a–h, j–m). The angles of the two **2a** molecules vary between 56.31° and 68.55° for the representative examples (Supplementary Table 10). Interestingly, a racemate of $\alpha$-$PC_{71}BM$ is found disordered with the occupancy of 0.5:0.5, while the V-shaped DPC hosts are observed to be fully ordered (Supplementary Fig. 10). For isomeric $C_s$-$C_{71}H_2$-I (Fig. 4h) and $C_{2v}$-$C_{71}H_2$-III (Fig. 4m) **2a** takes a V-shape conformation, but for $C_s$-$C_{71}H_2$-II (Fig. 4i) **2a** adopts an almost parallel sandwich-like conformation in the co-crystals. Additionally, for $Sc_3N@C_{80}$, the conformation of the 2DPC{$Sc_3N@C_{80}$} changes into sandwich-like structure (Fig. 4n). Interestingly, a sandwich-like conformation is found for 2DPC{$(C_{59}N)_2$} as well, in which the two **2a** molecules are dramatically dislocated so that each DPC host molecule can tightly bind one of the fullerene cages in $(C_{59}N)_2$ (Fig. 4o), further demonstrating the flexibility of the present DPC host **2a**.

Although the goal of sorting chiral fullerene enantiomers was not achieved in the example of 2DPC{$\alpha$-$PC_{71}BM$}, it is worth trying more experiments in the future. Sorting chiral fullerene enantiomers might be possible since DPC can become chiral upon crystallization though it is intrinsically achiral in solution, as mentioned in Balch's article[34]. This important characteristic provides valuable opportunities and new thoughts for sorting fullerene enantiomers by DPC.

The merit of **2a** is even further shown in the assembly of various fullerenes into two-dimensional (2D) layered structures. Benefiting from the flexible pyrryl groups, **2a** itself adopts a unique layer-by-layer packing mode that constructs potential untethered cavities for fullerene intercalation (Fig. 2b). Interestingly, in the supramolecular assembly of DPC host **2a** with any buckyball, the fullerene molecules could intercalate into the

cavities to form multilayered matrix structures composed of alternating fullerene mono layers and DPC double layers as outlined in Fig. 5 for $C_{60}$ and Supplementary Fig. 11 for the representative fullerene molecules (including $C_{70}$, $Sc_3N@C_{80}$, $\alpha$-$PC_{71}BM$ and $C_{2v}$-$C_{71}H_2$-III). In the multi-layered structure, the DPC layer adopts a slightly corrugated arrangement, but the fullerene layers form an interesting 2D molecular assembly with $C_{60}$ centroid-to-centroid separations of 14.23 Å and 14.12 Å along the $a$ axis and the $c$ axis, respectively (Fig. 5b), and an interlayer spacing of 16.93 Å (Fig. 5a). The 2D arrangement of metallofullerenes was reported as one special assembly example by Shinohara and co-workers[43]. Note that controlled fabrications of ordered 2D fullerene molecular nanostructures are challenging and mainly accomplished by means of metal surface-supported self-assembly approaches previously[44–47].

**Theoretical understanding of the assembly of DPC host/buckyball guest.** To understand the assembly between fullerenes and **2a**, DFT theoretical calculations were performed on DPC (**2a**) and the representative 2DPC{fullerene} co-crystals (Supplementary Note 8). Similar to the parent corannulene molecule, the lowest unoccupied molecular orbital (LUMO) and highest occupied molecular orbital (HOMO) of **2a** are both doubly degenerate[48]. However, the degenerate HOMO of **2a** is mainly localized on the pyrryl groups, while the LUMO is distributed on the corannulene core, and the HOMO-LUMO gap (2.80 eV) is lower than that of pristine corannulene (4.39 eV) at the B3LYP-D3BJ/6-31 G(d,p) level[49] (see Supplementary Fig. 17). The total interaction energies of 2DPC{fullerene} range from −62.60 to −76.54 kcal mol$^{-1}$, as presented in Fig. 6 and Table S13. Taking $C_{60}$ as an example, the total interaction energy, namely, $\Delta E^{int}_{F/2DPC}$, is −62.60 kcal mol$^{-1}$, which is much larger than that of the double concave buckycatcher reported by Sygula and co-workers[7] (−35.60 kcal mol$^{-1}$) at the same calculation level, see Supplementary Data 1).

In addition, according to fragmentation methods[50], **2a** can be divided into pyrryl fragments (PFs) and corannulene fragment (CF) by capping the dangling bonds with H atoms. As a result, the total interaction energy of 2DPC{fullerene}, namely, $\Delta E^{int}_{F/2DPC}$, can be divided into three parts, including the interaction energy between fullerene and twenty PFs ($\Delta E^{int}_{F/P}$), the interaction energy between fullerene and two CFs ($\Delta E^{int}_{F/C}$) and the two-body interaction correlation energy between PFs and CFs ($\Delta E^{corr}$). That is, $\Delta E^{int}_{F/2DPC} = \Delta E^{int}_{F/P} + \Delta E^{int}_{F/C} + \Delta E^{corr}$.

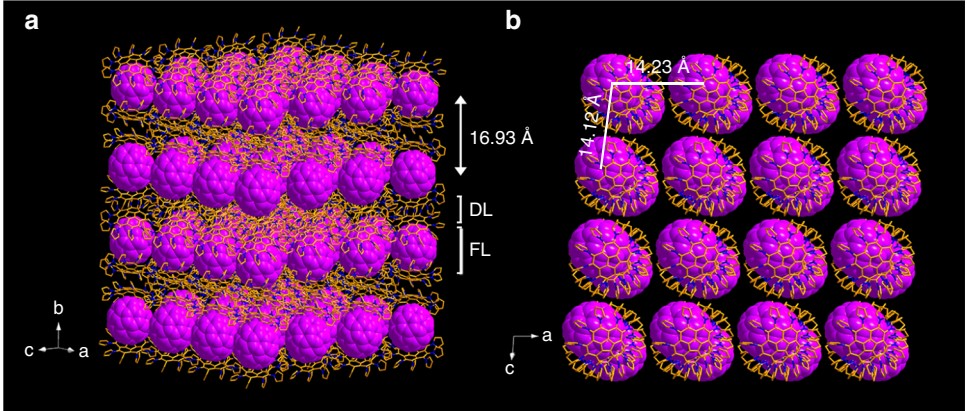

**Fig. 5** Layered structure of the co-crystal formed between DPC (**2a**) and $C_{60}$. **a, b** Edge-on view (**a**) and top view (**b**) of the space-filling structure of $C_{60}$ and the stick structure of DPC. FL is a monolayer of $C_{60}$, and DL is a slightly corrugated double DPC layer. C atom is shown in gold, N atom in navy blue, and $C_{60}$ in purple. All hydrogens and solvent molecules ($CS_2$) in the co-crystal have been omitted for clarity

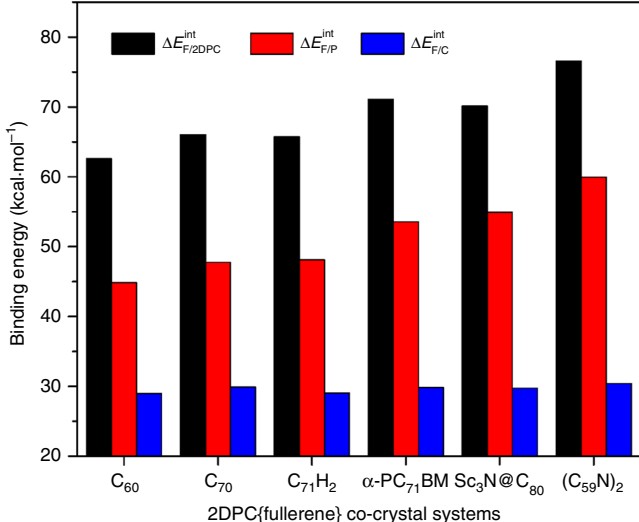

**Fig. 6** The total interaction energies and partial interaction energies for six representative co-crystals. All values in the figure are absolute values. $\Delta E^{int}_{F/2DPC}$ (black columns), $\Delta E^{int}_{F/P}$ (red columns), and $\Delta E^{int}_{F/C}$ (blue columns) represent the interaction energies between fullerene (F) and both hosts (2DPC), fullerene (F) and pyrryl groups (P), and fullerene (F) and corannulene part (C), respectively

As outlined in Fig. 6 and Supplementary Table 14, the partial interaction energies of PFs and fullerenes are generally 1.5–1.9 times larger than those of CF and fullerenes in the representative supramolecules, revealing that the pyrryl fragments make a larger contribution to the total interaction energy than the corannulene fragments ($\Delta E^{int}_{F/P} > \Delta E^{int}_{F/C}$). In addition, the $\Delta E^{int}_{F/2DPC}$ (black columns) increases with the $\Delta E^{int}_{F/P}$ (red columns), whereas the $\Delta E^{int}_{F/C}$ (blue columns) changes slightly, suggesting that the variation in the total interaction energies mainly depends on changes in the partial interaction energies of PFs and fullerene.

Accordingly, we have demonstrated that the DPC host molecule with a concave corannulene and flexible groups is able to adaptably hold various fullerenes to form (+)hand-ball-hand (−) supramolecular 2DPC{fullerene} assemblies, as exemplified by 15 typical fullerenes including pristine, endohedral, exohedral, dimeric and hetero-fullerene as well as fulleroid and non-IPR fullerene. Such a particular interaction between DPC and fullerene can be attributed to the electron-rich property, expanded and matched surface contact, and especially the flexible and self-adaptable pyrryl groups. DFT theoretical calculations confirm that the crucial contribution of the pyrryl groups is actually more than that of corannulene core in the supramolecular assemblies of 2DPC{fullerene}. Besides, benefitting from the flexible pyrryl groups and the concave corannulene, the DPC hosts themselves assemble into a layer-by-layer crystal, suitable for intercalation of various fullerenes as two-dimensional structures. In the future, expectedly, more fullerenes may be crystallographically identified based on the supramolecular (+) hand-ball-hand(−) 2DPC{fullerene} co-crystals, and the DPC host-decorated materials may be capable of separating buckyballs in more complex matrixes.

## Methods

**Synthesis and separation of compound 2.** To a DMF (15 ml) solution of NaH (60% oil dispersion, 80.7 mg, 2.00 mmol for **2a** or 136.0 mg, 3.40 mmol for **2b**) was added 135.1 mg (2.01 mmol) of pyrrol for **2a** or 323.0 mg (3.40 mmol) of 3,4-dimethylpyrrol for **2b** at 0 °C. After the evolution of $H_2$ gas ceased, the reaction mixture was stirred for an additional 30 min at the same temperature, and then,

decachlorocorannulene **1** (100.0 mg, 0.17 mmol) was added. The reaction mixture was stirred for an additional 12 h at 25 °C before it was poured into ice water (20 ml). Then, the mixture was added to dichloromethane (20 ml). The organic layers were separated, and the aqueous layer was thoroughly extracted with dichloromethane. The combined organic layers were washed with water and brine and dried over magnesium sulfate. After filtration and evaporation of the solvent, product **2a** was separated by an alkaline aluminium oxide column, and product **2b** was separated by silica gel column chromatography using dichloromethane/hexane (1:1) as the eluent. Finally, 45.3 mg (0.05 mmol) of **2a** or 40.1 mg (0.034 mmol) of **2b** was obtained. The isolated yield is approximately 30% for **2a** or 20 % for **2b**.

**X-ray diffraction analysis.** Slow evaporation from a solution of **2a** and fullerenes in toluene or a mixture of toluene and dichloromethane or a mixture of disulfide and dichloromethane provided co-crystals of 2DPC{fullerene} suitable for X-ray crystal structure analysis (see Supplementary Note 5 and Supplementary Table 2 for detail). In each case, a suitable crystal was mounted with mineral oil on a glass fibre and transferred to an Agilent SuperNova diffractometer with a Cu $K\alpha$ ($\lambda = 1.54184$ Å) microfocus X-ray source. The data were processed using CrysAlis$^{Pro}$. The structure was solved and refined using full-matrix least-squares based on $F^2$ with the programs SHELXT and SHELXL-2015[51] within OLEX2[52]. The intensities were corrected for Lorentz and polarization effects. The non-hydrogen atoms were refined anisotropically. Hydrogen atoms were placed using AFIX instructions. Program SQUEEZE, a part of the PLATON package[53] of crystallographic software, was used to calculate the solvent disorder area and remove its contribution from the intensity data if needed.

## Data availability

The data that support the findings of this study are available from the corresponding author upon request. The crystallographic data for the structures reported in this paper have been deposited at the Cambridge Crystallographic Data Centre (CCDC) under the deposition numbers CCDC 1827393 [DPC (**2a**), $C_{60}H_{40}N_{10}$], CCDC 1827400 [DPC (**2b**), $C_{80}H_{80}N_{10}$], CCDC 1827417 [2DPC{$C_{60}$}, $C_{180}H_{80}N_{20}$], CCDC 1827406 [2DPC{$C_{70}$}, $C_{190}H_{80}N_{20}$], CCDC 1856253 [2DPC {$C_{90}$}, $C_{210}H_{80}N_{20}$], CCDC 1827404 [2DPC{$\alpha$-PC$_{71}$BM}, $C_{202}H_{94}N_{20}O_2$], CCDC 1856254 [2DPC{$\beta_1$-PC$_{71}$BM}, $C_{202}H_{94}N_{20}O_2$], CCDC 1856255 [2DPC{PC$_{61}$P}, $C_{192}H_{89}N_{21}$], CCDC 1856256 [2DPC{PC$_{61}$AE}, $C_{190}H_{91}N_{21}O_2$], CCDC 1856262 [2DPC{$C_{71}$H$_2$-I}, $C_{191}H_{82}N_{20}$], CCDC 1856263 [2DPC{$C_{71}$H$_2$-II}, $C_{191}H_{82}N_{20}$], CCDC 1856383 [2DPC{$C_{65}$H$_6$}, $C_{185}H_{86}N_{20}$], CCDC 1856270 [2DPC{$C_{60}$HPh}, $C_{186}H_{86}N_{20}$], CCDC 1856271 [2DPC{$C_{60}$HCH$_3$}, $C_{186}H_{86}N_{20}$], CCDC 1827408 [2DPC{$C_{71}$H$_2$-III}, $C_{191}H_{82}N_{20}$], CCDC 1827407 [2DPC{Sc$_3$N@C$_{80}$}, $C_{200}H_{80}N_{21}Sc_3$] and CCDC 1827410 [2DPC{($C_{59}$N)$_2$}, $C_{238}H_{80}N_{22}$]. Copies of the data can be obtained free of charge from [www.ccdc.cam.ac.uk/data_request/cif].

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

## Acknowledgements

This research was supported by the National Natural Science Foundation of China (21771152, 21721001, 21827801, 51572231, 51572254, 21571151, 2170010228), the 973 Program of China (2014CB845601 and 2015CB932301), the China Postdoctoral Science Foundation (2016M602067), the National Key Research and Development Program of China (2017YFA0402800), and the Fundamental Research Funds for the Central Universities (20720170028, 20720160084). Q.Y.Z. is particularly grateful to 21771152, 2015CB932301, 20720170028, 20720160084; S.F.Y. is particularly grateful to 51572254 and 2017YFA0402800; S.Y.X. is particularly grateful to 21721001 and 51572231; L.S.Z. is particularly grateful to 21827801; S.L.D. is particularly grateful to 21571151; S.H.L. is particularly grateful to 2170010228 and 2016M602067.

## Author contributions

Y.Y.X. conducted organic synthesis and structural identification of the DPC molecule, and assembled DPC with $C_{60}$ and $C_{70}$, as well as participated in drafting the manuscript. H.R.T. performed X-ray crystallographic measurements, isolated and crystallized $C_s$-$C_{71}H_2$-I, $C_s$-$C_{71}H_2$-II, $C_{2v}$-$C_{71}H_2$-III and non-IPR $C_{65}H_6$ from combustion soot, and solved some of the structures. S.H.L. synthesized and crystallized the derivatives of [6,6]-phenyl-$C_{61}$-pyridyl and [6,6]-pyridyl-$C_{61}$-acetic acid ethyl ester. Z.C.C. performed the theoretical calculations. Y.R.Y. solved the crystallographic structures and prepared CIF files. S.S.W. synthesized the derivatives of $C_{60}$HPh and $C_{60}$HCH$_3$. X.Z. separated and crystallized α- and β$_1$-PC$_{71}$BM isomers. Z.Z.Z. synthesized $(C_{59}N)_2$. S.L.D. participated in designing the experiment and analyzing the data. Q.Y.Z. designed the DPC molecule, proposed the synthetic scheme, guided the experiments and co-wrote the manuscript. S.F.Y. afforded $C_{90}$ and Sc$_3$N@$C_{80}$, and participated in analyzing the data and revising the manuscript. S.Y.X. conceived the research, analyzed the data and revised the manuscript. R.B.H. participated in analyzing the data. L.S.Z. participated in analyzing the data. All the authors discussed the data and commented on the manuscript.

## Additional information

**Competing interests:** The authors declare no competing interests.

