## [Peer Review File · Nature Communications]

Reviewers' comments:

Reviewer #1 (Remarks to the Author):

This manuscript describes the synthesis of a unique corannulene, which shows generality for the cocrystallization with various(15) kinds of fullerenes, i.e. pristine, exohedral, endohedral and hetero-derivatized. The study of fullerenes and their derivatives have been long suffered from the difficulties of their structural characterization. The X-ray single crystal analysis has been the most reliable method so far to determine the fine structure of fullerenes. However, few compound has been proved to be able to cocrystallize with various forms of fullerenes, especially with endohedral fullerenes and fullerene derivatives. In this work, the authors demonstrate that decapyrrylcorannulene can efficiently cocrystallize with all kinds of known forms of fullerenes, thus solved some of fullerene structures which have never been obtained before. In addition, the mechanism of proposed 'buckyhand' concept has been well interpreted by DFT calculation results. This finding is expected to largely facilitate the structure determination of novel fullerene structures and the strategy of 'buckyhand' may also be further developed and applied to supramolecular assembly and crystallization of other molecules with spherical 3D structures. This manuscript was well written and I believe in general it reaches requirements of originality, significance and impact of Nature Communication. Thus I am looking favorable on this contribution and recommend the final acceptance of this manuscript after the author addressed the following questions:

1. The authors have synthesized both compound 2a and 2b to compare and did mention that 2b is not as efficient as 2a when cocrystallized with fullerenes, but very few experimental data has been provided for the possible supermolecular interaction between 2b and variable fullerenes. Did 2b do not interact with all the fullerenes at all or it works with some of them? How about the theoretical binding energy between 2b and C60 (or other fullerenes mentioned)? I suggest that the authors provide some detailed data for 2b, as it is important for the understanding the uniqueness of 2a and its interaction with variable fullerenes through this comparison.

2 The authors gave a very brief description of the preparation of cocrystals of 2DPC(fullerene) in Methods. A more detailed experimental methods of the preparation of 15 co-crystals should be given in the supporting information, as experimental conditions, such as solvents, seem to vary from case to case and these details are essential for the reproduction of this work.

3. How about the stability of these DPC(2a)-fullerene cocrystals? Is it possible to decomplexate the cocrystal and recycle the fullerenes under certain condition?

4 Crystallographic analysis of endohedral fullerenes has been long suffered by the disorder of encapsulated clusters or ions in most of the fullerene-Ni(II)OEP cocrystal. DPC(2a) seems have a stronger and different supramolecular interaction with fullerenes, does this interaction also reduce the disorder of the encapsulated species?

5 In page 1, line 55, 'Tens fullerenes' seems to be a typo. It's likely to be 'Tens of fullerenes', please correct or clarify this.

6 In page 9, line 305, another two important work about the supramolecular self-assembly 2D fullerene structure: Pan GB et al., J. Am. Chem. Soc. 2006,128, 13, 4218-421; Wan, LJ, et al Angew. Chem. Int. Ed. 2003, 42, 2747 – 2751, could be cited together with reference 43 and 44.

Reviewer #2 (Remarks to the Author):

This is an interesting manuscript which describes the synthesis of two new decapyrrylcorannulene having a concave surface of corannulene and ten flexible electron-rich units of pyrrolyl group to mimic the so-called by the authors molecular 'buckyhands'. Although I personally do not like this kind of names out of the chemical aspects, this is a decision of authors.

The new molecules are not totally new in the sense that previously the related molecule formed by a corannulene decorated with ten phenyl groups was reported (ref. 30). However, the main merit of this manuscript is mostly in the use of the new molecule for crystallising a variety of 15 fullerenes (including pristine, endohedral, exohedral, dimeric and hetero-fullerene as well as fulleroid and non-IPR fullerene), whose X-ray diffractions have been nicely described.

The aforementioned crystallinity stems from the electron-rich pyrrolyl nature as well as the surface contact and, especially, from the flexible pyrrolyl units. DFT theoretical calculations predict the critical contribution of the pyrrolyl units in the supramolecular assemblies, significantly stronger than that of the corannulene core. As a whole, this is a not new but useful alternative to the use of, for instance, porphyrins for crystallising and identifying new and emergent fullerenes and, in particular, endohedrals.

The work has been competently carried out and it is clearly presented. Therefore, I feel it meets the requirements of novelty and quality to be accepted for publication in Nature Communications after addressing the following minor points:

1. Figure S-7 showing the electronic properties of the new molecules should be explained more in detail. Actually, according to DFT calculations, a charge transfer band should be observed and nothing is mentioned about the electronic properties of the molecules synthesized. Please, clarify the nature of the observed bands...

Reviewer #3 (Remarks to the Author):

A rather large number of receptors for fullerene have been made. Generally, such studies involve a significant effort in synthetic chemistry and some effort to demonstrate binding either spectroscopically or crystallographically. Frequently, the receptor is never utilized again. Here another receptor is synthesized, but in this case it may have some extended utility if it can become readily available.

This paper describes a new receptor (DPC) that is useful in crystallization of fullerenes for crystallography. The data in Figure 4 are impressive. Fifteen different fullerenes over a distribution of sizes and shapes have been found to crystallize with DPC. This Figure shows both the strength of the article (many fullerenes bind DPC) and its weaknesses (most of the individual structures are given too little attention).

The use of DPC to obtain a crystal of the dimer (C₅₉N)₂ is particularly significant. Indeed, this structure should receive more attention in the paper. Where is the nitrogen atom? What is the length of the bond connecting the two fullerenes?

The authors have overlooked an aspect of the buckyhands they have made that may be quite interesting and useful. Since the receptor is itself achiral but can become chiral upon crystallization, it may be useful in crystallizing chiral fullerenes and sorting the two hands into individual crystalline sites. Currently, most chiral fullerene crystallize in a disordered fashion. Sorting them into individual chiral sites in crystals, without actual bulk resolution, could be a useful structural technique. The concept can be found in Scheme 1 of the following article, which the authors should cite: Ghiassi, K. B., et al. *Crystal Growth & Design* 2016, 16: 447-455. A copy of Scheme 1 is attached.

The 15 structures shown in Figure 4 vary in quality. Some are ordered, others have disorder in the fullerene, in the solvates involved, and even in the DPC. I have no problem accepting that structures are disordered, but I would like the disorder described. Thus, the Supporting

Information should present a careful discussion of the disorder in each of these 15 cocrystals.

The source of the decachlorocorannulene should be given. How readily is this starting material available? Its availability may determine whether others will be able to utilize DPC in other capacities.

Alan L. Balch

Reviewer #1 (Remarks to the Author):

This manuscript describes the synthesis of a unique corannulene, which shows generality for the cocrystallization with various (15) kinds of fullerenes, i.e. pristine, exohedral, endohedral and hetero-derivatized. The study of fullerenes and their derivatives have been long suffered from the difficulties of their structural characterization. The X-ray single crystal analysis has been the most reliable method so far to determine the fine structure of fullerenes. However, few compound has been proved to be able to cocrystallize with various forms of fullerenes, especially with endohedral fullerenes and fullerene derivatives. In this work, the authors demonstrate that decapyrrylcorannulene can efficiently cocrystallize with all kinds of known forms of fullerenes, thus solved some of fullerene structures which have never been obtained before. In addition, the mechanism of proposed ‘buckyhand’ concept has been well interpreted by DFT calculation results. This finding is expected to largely facilitate the structure determination of novel fullerene structures and the strategy of ‘buckyhand’ may also be further developed and applied to supramolecular assembly and crystallization of other molecules with spherical 3D structures. This manuscript was well written and I believe in general it reaches requirements of originality, significance and impact of Nature Communication. Thus I am looking favorable on this contribution and recommend the final acceptance of this manuscript after the author addressed the following questions:

1. The authors have synthesized both compound **2a** and **2b** to compare and did mention that **2b** is not as efficient as **2a** when cocrystallized with fullerenes, but very few experimental data has been provided for the possible supermolecular interaction between **2b** and variable fullerenes. Did **2b** do not interact with all the fullerenes at all or it works with some of them? How about the theoretical binding energy between **2b** and C₆₀ (or other fullerenes mentioned)? I suggest that the authors provide some detailed data for **2b**, as it is important for the understanding the uniqueness of **2a** and its interaction with variable fullerenes through this comparison.

Response: Thank the reviewer for the comment and suggestion, which embodies the uniqueness of **2a** in binding to fullerenes. In the revised supplementary information, the theoretical binding energy between **2b** and C₆₀ was calculated, revealing an interaction of **2b** - C₆₀ slightly stronger than that of **2a** - C₆₀. Just because β-H of pyrryl group in **2a** is replaced by methyl group, however, the intermolecular interaction of **2b** - **2b** is much stronger than that of **2a** - **2a**. Consequently, we obtained single crystals of **2b** itself but failed to get the co-crystal of **2b** and fullerene like C₆₀. To better understand the uniqueness, three molecular stacking models of **2a** and **2b** were optimized respectively. In the three molecular packing of **2a**, the cavity suitable for fullerenes entering can be theoretically predicted, complying with the experimental observation in the crystal packing of **2a**. Whereas in the three molecular packing of **2b**, the key cavity is found to disappear. On the other hand, the

interactions of **2b** itself are much larger than that of **2b** and C₆₀, as a result, **2b** is inclined to self-assemble rather than assemble with fullerenes according to theoretical prediction. Consequently, these critical factors possibly lead to the failure of co-crystal between **2b** and fullerenes. We also attempted to get the optimized four molecular packing of **2a** and **2b**, but the system is too big for us to calculate at the present time. The detailed calculation data for three molecular packings of **2a** and **2b** has been provided in Fig. S19 and Table S15 of the revised Supplementary Information.

2 The authors gave a very brief description of the preparation of cocrystals of 2DPC(fullerene) in Methods. A more detailed experimental methods of the preparation of 15 co-crystals should be given in the supporting information, as experimental conditions, such as solvents, seem to vary from case to case and these details are essential for the reproduction of this work.

Response: Thank the reviewer for the suggestion. We realize the missing detailed conditions for all the 15 co-crystals. Now a new Table S2 and text describing detailed experimental methods about growth of all the 15 co-crystals have been added in the section IV Crystal Information of revised Supplementary Information. We believe all the experiments can be simply repeated by other groups.

3. How about the stability of these DPC(2a)-fullerene cocrystals? Is it possible to decomplexate the cocrystal and recycle the fullerenes under certain condition?

Response: The DPC (**2a**)-fullerene co-crystals are stable for several weeks under ambient air at room temperature. For the co-crystals with the involvement of carbon disulfide, both the fullerenes and DPC (**2a**) are recyclable with recoveries more than 90% through ultrasonic decomplexation followed by column chromatography separation or HPLC separation. For the toluene-involving co-crystals, both the DPC (2a) and fullerenes can be recycled but with lower recovery. The description of stability and recycling are added in the crystal information (IV) of the revised Supplementary Information.

4. Crystallographic analysis of endohedral fullerenes has been long suffered by the disorder of encapsulated clusters or ions in most of the fullerene-Ni(II)OEP cocrystal. DPC(2a) seems have a stronger and different supramolecular interaction with fullerenes, does this interaction also reduce the disorder of the encapsulated species?

Response: As the reviewer mentioned, crystallographic analysis of endohedral fullerenes has been long suffered by the disorder of encapsulated species or fullerene cage, which is absolutely a complicated problem and affected by many factors during crystals growth, such as temperature, solvent, the nature of host, as well as the encapsulated cluster itself. DPC (**2a**) displays a strong supramolecular

interaction with endohedral fullerenes, as exemplified in the co-crystals of DPC (**2a**)-Sc₃N@C₈₀. Sc₃N@C₈₀, whose molecular structure was previously determined in a co-crystal with Co^{II}(OEP) but suffered from serious disorder defects (*Nature* 1999, 401, 55-57), while in the co-crystal of DPC (**2a**)-Sc₃N@C₈₀, the C₈₀ cage is basically ordered though the encapsulated scandium atom is disordered. From our viewpoint, the encapsulated species seem to be less affected by the DPC host, largely because the encapsulated species are shielded by the fullerene cage and they seem to be more affected by the cage. However, it may be still too early to make the conclusion due to lack of more examples at this stage.

5 In page 1, line 55, 'Tens fullerenes' seems to be a typo. It's likely to be 'Tens of fullerenes', please correct or clarify this.

Response: Thank the reviewer for the carefulness. It should be a typo. We have corrected it in the revised manuscript according to the reviewer's suggestion.

6 In page 9, line 305, another two important work about the supramolecular self-assembly 2D fullerene structure: Pan GB et al., *J. Am. Chem. Soc.* 2006,128, 13, 4218-421; Wan, LJ, et al *Angew. Chem. Int. Ed.* 2003, 42, 2747 - 2751, could be cited together with reference 43 and 44.

Response: Thank the reviewer for the suggestion. We do realize that those two works recommended by the reviewer have a similar importance with references 43 and 44. Considering the reference limitation (normally less than 50), we hope that editors will agree to increase the number of references in the revision for additionally citing those two references as references 46 and 47 (*J. Am. Chem. Soc.* 2006,128, 13, 4218-421; *Angew. Chem. Int. Ed.* 2003, 42, 2747-2751).

Reviewer #2 (Remarks to the Author):

This is an interesting manuscript which describes the synthesis of two new decapyrrylcorannulene having a concave surface of corannulene and ten flexible electron-rich units of pyrryl group to mimic the so-called by the authors molecular 'buckyhands'. Although I personally do not like this kind of names out of the chemical aspects, this is a decision of authors.

Response: We would like to thank the reviewer for his/her comment. We agree with the reviewer that the name of molecular 'DPC buckyhands' is somewhat out of the chemical aspects. In the revised manuscript and supporting information, we have used a more chemically descriptive name ('DPC host') instead of the previous name ('DPC buckyhands'). Besides, the 'fingers' has also been changed to flexible pyrryl groups in the revised manuscript and supporting information.

The new molecules are not totally new in the sense that previously the related molecule formed by a corannulene decorated with ten phenyl groups was reported (ref. 30). However, the main merit of this manuscript is mostly in the use of the new molecule for crystallising a variety of 15 fullerenes (including pristine, endohedral, exohedral, dimeric and hetero-fullerene as well as fulleroid and non-IPR fullerene), whose X-ray diffractions have been nicely described.

The aforementioned crystallinity stems from the electron-rich pyrrol nature as well as the surface contact and, especially, from the flexible pyrrol units. DFT theoretical calculations predict the critical contribution of the pyrrol units in the supramolecular assemblies, significantly stronger than that of the corannulene core. As a whole, this is a not new but useful alternative to the use of, for instance, porphyrins for crystallising and identificate new and emergent fullerenes and, in particular, endoedrals.

The work has been competently carried out and it is clearly presented. Therefore, I feel it meets the requirements of novelty and quality to be accepted for publication in Nature Communications after addressing the following minor points:

1. Figure S-7 showing the electronic properties of the new molecules should be explained more in detail. Actually, according to DFT calculations, a charge transfer band should be observed and nothing is mentioned about the electronic properties of the molecules synthesized. Please, clarify the nature of the observed bands...

Response: Thank the reviewer for his/her constructive comment and suggestion. A charge transfer band has indeed been observed at around 383 nm as predicted. Time dependent (TD) DFT calculation result of **2a** molecule at the B3LYP-D3BJ/6-31G(d) level showing a charge transfer band at $\lambda = 383.79$ nm and $\lambda = 383.75$ nm are assigned to HOMO-13/HOMO-14 \rightarrow LUMO/LUMO+1 and HOMO-15/HOMO-16 \rightarrow LUMO/LUMO+1 transitions. The detailed discussion about the observed electronic properties have now been added at section III Electronic Properties of DPC (**2a**), and the experimental and computational data shown in the new Figs. S7-8 and Table S1 of the revised Supplementary Information.

Reviewer #3 (Remarks to the Author):

A rather large number of receptors for fullerene have been made. Generally, such studies involve a significant effort in synthetic chemistry and some effort to demonstrate binding either spectroscopically or crystallographically. Frequently, the receptor is never utilized again. Here another receptor is synthesized, but in this case it may have some extended utility if it can become readily available.

This paper describes a new receptor (DPC) that is useful in crystallization of fullerenes for crystallography. The data in Figure 4 are impressive. Fifteen different

fullerenes over a distribution of sizes and shapes have been found to crystallize with DPC. This Figure shows both the strength of the article (many fullerenes bind DPC) and its weaknesses (most of the individual structures are given too little attention).

The use of DPC to obtain a crystal of the dimer $(C_{59}N)_2$ is particularly significant. Indeed, this structure should receive more attention in the paper. Where is the nitrogen atom? What is the length of the bond connecting the two fullerenes?

Response: Thank the reviewer for his constructive comment and suggestion. A total of 15 structures having a variety of complex structural features deserve to pay more attention, but only a few were selected to discuss in the original manuscript. For example, the fulleriod of C_{2v} - $C_{71}H_2$ -III and its isomers C_s - $C_{71}H_2$ -I and C_s - $C_{71}H_2$ -II have been discussed in the text of the manuscript and the Supplementary Figs S5-6 in details. For the $(C_{59}N)_2$ we do agree the reviewer that the structure should receive more attention, but the crystal of $(C_{59}N)_2$ suffered a slight disorder. The disorder might be due to the small amount of anti-conformer in the equilibrium mixtures of anti and gauche-conformers predicted in the reference 41 (*J. Am. Chem. Soc.* 2001, 123, 11085-11086). Due to low occupancy (about 10-15%), the occupancy of the major site is refined to 100% with some DFIX commands to regulate the cage. Accordingly, the lengths of the bonds in $(C_{59}N)_2$ are too inaccurate to discuss in the present work. In this case, the nitrogen atom is reasonably determined according to the previous literature (reference 37: *Science* 1995, 269, 1554-1556), in which the exact structure of $(C_{59}N)_2$ has been identified according to its' two planes of symmetry (or one inversion center and one plane of symmetry) as inferred from only 30 lines in the ^{13}C -NMR spectrum of $(C_{59}N)_2$.

The authors have overlooked an aspect of the buckyhands they have made that may be quite interesting and useful. Since the receptor is itself achiral but can become chiral upon crystallization, it may be useful in crystallizing chiral fullerenes and sorting the two hands into individual crystalline sites. Currently, most chiral fullerene crystallize in a disordered fashion. Sorting them into individual chiral sites in crystals, without actual bulk resolution, could be a useful structural technique. The concept can be found in Scheme 1 of the following article, which the authors should cite: Ghiassi, K. B., et al. *Crystal Growth & Design* 2016, 16: 447-455. A copy of Scheme 1 is attached.

Response: Thank the reviewer for such a constructive suggestion. We do agree that it is a very important and interesting concept to sort fullerene enantiomers into individual chiral sites in co-crystals as reported in the article (Ghiassi, K. B., et al. *Crystal Growth & Design* 2016, 16: 447-455). Based on the concept, we have paid special attention to scrutinize all the co-crystals. However, many of the selected fullerenes are achiral. In our revision, the co-crystal of DPC- α -PC $_{71}$ BM is rethought further. Interestingly, a racemate of α -PC $_{71}$ BM is found disordered with the occupancy of 0.5:0.5 between a pair of DPC hosts, while the pair of DPC hosts are

observed to be fully ordered (shown in the new Fig. S10 in the Supplementary Information). Although it is too early to conclude the goal of sorting chiral fullerenes, it is worth of more experiments in the future. As mentioned, the DPC can become chiral upon crystallization though it is intrinsically achiral in solution, the important characteristic provides valuable opportunities and new thoughts for sorting fullerene enantiomers by DPC. We have stated this point in the text of the revised manuscript (in the section of supramolecular behaviours of the DPC host/buckyball guest systems). In addition, we hope that editors will agree to increase the article reported by Prof. Balch (*Crystal Growth & Design* 2016, 16: 447-455) as reference 34 in the revised references.

The 15 structures shown in Figure 4 vary in quality. Some are ordered, others have disorder in the fullerene, in the solvates involved, and even in the DPC. I have no problem accepting that structures are disordered, but I would like the disorder described. Thus, the Supporting Information should present a careful discussion of the disorder in each of these 15 cocrystals.

Response: Thank the reviewer for the suggestion. The new Table S8 containing detailed descriptions of all disorders in each of the 15 co-crystals was presented in the revised supporting information.

The source of the decachlorocorannulene should be given. How readily is this starting material available? Its availability may determine whether others will be able to utilize DPC in other capacities.

Response: Thank the reviewer for the kind suggestion. Decachlorocorannulene can be simply synthesized by one step of perchlorination reaction as reported by Sigel et al. in the article of *J. Am. Chem. Soc.* **121**, 7439-7440 (1999), which has been cited as a new reference 1 in the revised Supplementary Information for explanation of the source of the decachlorocorannulene. The original reference 1 in the Supplementary Information has been deleted because which was found to be duplicated with reference 40 in the manuscript.

In addition, according to our recent reanalysis of single crystal structures, the original alert level A in co-crystal of 2DPC{C₆₀HCH₃} has been eliminated now. In this case, we updated the CIF document of 2DPC{C₆₀HCH₃} and the related crystal information of Table S7 in the revised Supplementary Information.

REVIEWERS' COMMENTS:

Reviewer #1 (Remarks to the Author):

I think the authors have properly addressed my questions and revised the ms and SI files accordingly. Thus I suggest this article can now be accepted as it is.

Reviewer #2 (Remarks to the Author):

The authors have nicely addressed the concerns raised not only by this reviewer but also of the remaining referees and, therefore, I feel that the manuscript in its present form meets the criteria of quality and novelty to be accepted for publication basically as it stands.

Reviewer #3 (Remarks to the Author):

The authors have provided well reasoned responses to each of the issues I raised in my review. I recommend publication of the article in its present form. Somewhere in the text the author's wrote 4o when they meant 40.

Reviewer #1 (Remarks to the Author):

I think the authors have properly addressed my questions and revised the ms and SI files accordingly. Thus I suggest this article can now be accepted as it is.

Response: Thank the reviewer very much for the acceptance of our responses and suggestion to publish in the present form.

Reviewer #2 (Remarks to the Author):

The authors have nicely addressed the concerns raised not only by this reviewer but also of the remaining referees and, therefore, I feel that the manuscript in its present form meets the criteria of quality and novelty to be accepted for publication basically as it stands.

Response: Thank the reviewer very much for the acceptance of our responses and suggestion to publish in the present form.

Reviewer #3 (Remarks to the Author):

The authors have provided well reasoned responses to each of the issues I raised in my review. I recommend publication of the article in its present form. Somewhere in the text the author's wrote 4o when they meant 40.

Response: Thank the reviewer very much for the acceptance of our responses and suggestion to publish. According to the reviewer's comment, we recheck all 4o and 40 in the manuscript, and ensure all 4o and 40 represent what they should mean.